# Active contraction of microtubule networks

**Peter J Foster[1]\*, Sebastian Fürthauer[2,3], Michael J Shelley[2], Daniel J Needleman[1,3]**

[1]John A. Paulson School of Engineering and Applied Sciences, FAS Center for Systems Biology, Harvard University, Cambridge, United States; [2]Courant Institute of Mathematical Science, New York University, New York, United States; [3]Department of Molecular and Cellular Biology, Harvard University, Cambridge, United States

**Abstract** Many cellular processes are driven by cytoskeletal assemblies. It remains unclear how cytoskeletal filaments and motor proteins organize into cellular scale structures and how molecular properties of cytoskeletal components affect the large-scale behaviors of these systems. Here, we investigate the self-organization of stabilized microtubules in *Xenopus* oocyte extracts and find that they can form macroscopic networks that spontaneously contract. We propose that these contractions are driven by the clustering of microtubule minus ends by dynein. Based on this idea, we construct an active fluid theory of network contractions, which predicts a dependence of the timescale of contraction on initial network geometry, a development of density inhomogeneities during contraction, a constant final network density, and a strong influence of dynein inhibition on the rate of contraction, all in quantitative agreement with experiments. These results demonstrate that the motor-driven clustering of filament ends is a generic mechanism leading to contraction.

**\*For correspondence:**
peterfoster@fas.harvard.edu

**Competing interests:** The authors declare that no competing interests exist.

## Introduction

The mechanics, motions, and internal organization of eukaryotic cells are largely determined by the cytoskeleton. The cytoskeleton consists of filaments, such as actin and microtubules, and molecular motors, which consume chemical energy to exert forces on and arrange the filaments into large-scale networks. Motor proteins, including dynein and roughly 14 different families of kinesin (*Wordeman, 2010*), organize microtubules to form the spindle, which segregates chromosomes during cell division. The motor protein myosin organizes actin filaments into networks which drive cell motility, polarity, cytokinesis, and left-right symmetry breakage (*Mitchinson and Cramer, 1996*; *Mayer et al., 2010*; *Naganathan et al., 2014*). The non-equilibrium nature of motor activity is essential for the organization of the cytoskeleton into these diverse sub-cellular structures, but it remains unclear how the interactions between filaments, different motor proteins, and other biomolecules influence the behaviors of the networks they form. In particular, it is difficult to extrapolate from the biochemical properties of motors characterized in reconstituted systems to the biological function of those motors *in vivo*. To address this question, we study self-organization of cytoskeletal filaments in *Xenopus* extracts, which recapitulate the biochemical complexity of the *in vivo* system.

The self-organization of cytoskeletal filaments has been extensively studied in cell extracts and in reconstituted systems of purified components. Actin can form macroscopic networks that exhibit a myosin-dependent bulk contraction (*Murrell and Gardel, 2012*; *Bendix et al., 2008*; *Köhler and Bausch, 2012*; *Alvarado et al., 2013*; *Szent-Györgyi, 1943*). Microtubule networks purified from neuronal extracts have also been observed to undergo bulk contraction (*Weisenberg and Cianci, 1984*), while microtubules in mitotic and meiotic extracts are found to assemble into asters

**eLife digest** The ability of cells to move, divide, and carry out other processes depends on networks of protein filaments and motor proteins collectively known as the cytoskeleton. The motor proteins can move along the filaments to transport molecules and larger structures around the cell, or to rearrange the filaments themselves.

The cytoskeleton of animal, plant, and other eukaryotic cells contains two main types of filaments, known as actin filaments and microtubules. Both types of filament have distinct ends, known as the plus and minus ends. Previous studies have revealed that networks of actin filaments can rapidly contract to drive the movement of muscles and other processes. However, it is not known whether networks of microtubules can also contract.

Foster et al. studied the microtubules in extracts made from the eggs of a frog called *Xenopus laevis*. The experiments show that these microtubules form networks that can spontaneously contract. Foster et al. propose that this contraction is caused by the minus ends of the microtubules clustering together due to the activities of a motor protein called dynein.

To test this idea, Foster et al. developed a mathematical model based on an 'active fluid' theory. This model makes predictions that agree very well with the experimental data. The next step in this work is to find out if this model of microtubule contraction applies to other networks of microtubules.

(*Gaglio et al., 1995*; *Mountain et al., 1999*; *Verde et al., 1991*). Aster formation in meiotic *Xenopus* egg extracts is dynein-dependent, and has been proposed to be driven by the clustering of microtubule minus ends by dynein (*Verde et al., 1991*). It has also been suggested that dynein binds to the minus ends of microtubules in spindles and clusters the minus ends of microtubules to form spindle poles (*Heald et al., 1996*; *Burbank et al., 2007*; *Khodjakov et al., 2003*; *Goshima et al., 2005*; *Elting et al., 2014*) and dynein has been shown to accumulate on microtubule minus ends in a purified system (*McKenney et al., 2014*). Purified solutions of microtubules and kinesin can also form asters (*Nédélec et al., 1997*; *Hentrich and Surrey, 2010*; *Urrutia et al., 1991*), or under other conditions, dynamic liquid crystalline networks (*Sanchez et al., 2012*). Hydrodynamic theories have been proposed to describe the behaviors of cytoskeletal networks on length scales that are much greater than the size of individual filaments and motor proteins (*Prost et al., 2015*, *Marchetti et al., 2013*). These phenomenological theories are based on symmetries and general principles of non-equilibrium physics, with the details of the microscopic process captured by a small number of effective parameters. As hydrodynamic theories are formulated at the continuum level, they cannot be used to derive the values of their associated parameters, which must be obtained from more microscopic theories (*Prost et al., 2015*, *Marchetti et al., 2013*) or by comparison to experiments (*Mayer et al., 2010*; *Brugués and Needleman, 2014*).

A key feature of networks of cytoskeletal filaments and motor proteins that enters hydrodynamic theories, and differentiates these non-equilibrium systems from passive polymer networks, is the presence of additional, active stresses (*Prost et al., 2015*, *Marchetti et al., 2013*). These active stresses can be contractile or extensile, with profound implications for the large-scale behavior of cytoskeletal networks. Contractile stresses can result from a preferred association of motors with filament ends (*Kruse and Jülicher, 2000*; *Hyman and Karsenti, 1996*), nonlinear elasticity of the network (*Liverpool et al., 2009*), or the buckling of individual filaments (*Murrell and Gardel, 2012*; *Lenz, 2014*; *Soares e Silva et al., 2011*). Extensile active stresses can arise from polarity sorting or result from the mechanical properties of individual molecular motors (*Gao et al., 2015*; *Blackwell et al. 2015*). In networks with dynamically growing and shrinking filaments, polymerization dynamics can also contribute to the active stress. Experimentally, acto-myosin systems (*Murrell and Gardel, 2012*; *Bendix et al., 2008*; *Köhler and Bausch, 2012*; *Alvarado et al., 2013*; *Szent-Györgyi, 1943*) and microtubule networks from neuronal extracts (*Weisenberg and Cianci, 1984*) are observed to be contractile, while purified solutions of microtubules and kinesin can form extensile liquid crystalline networks (*Sanchez et al., 2012*). It is unclear which microscopic properties of

filaments and motor proteins dictate if the active stress is contractile or extensile in these different systems.

Here, we investigate the motor-driven self-organization of stabilized microtubules in *Xenopus* meiotic egg extracts. These extracts are nearly undiluted cytoplasm and recapitulate a range of cell biological processes, including spindle assembly and chromosome segregation (*Hannak and Heald, 2006*). We have discovered that, in addition to microtubules forming asters in this system as previously reported (*Verde et al., 1991*), the asters assemble themselves into a macroscopic network that undergoes a bulk contraction. We quantitatively characterized these contractions and found that their detailed behavior can be well understood using a simple coarse-grained model of a microtubule network in which dynein drives the clustering of microtubule minus ends. This end clustering mechanism leads to a novel form of active stress, which drives the system to a preferred microtubule density. Our results suggest that the dynein-driven clustering of microtubule minus ends causes both aster formation and network contraction, and have strong implications for understanding the role of dynein in spindle assembly and pole formation. Furthermore, the close agreement we find between experiments and theory demonstrates that simple continuum models can accurately describe the behavior of the cytoskeleton, even in complex biological systems.

## Results

To further study the motor-induced organization of microtubules, we added 2.5 $\mu$M Taxol to *Xenopus* egg extracts and loaded them into microfluidic channels (*Figure 1A*). Taxol causes microtubules to rapidly assemble and stabilize (*Mitchison et al., 2013*), which allowed us to decouple the effects of motor-driven self-assembly from the complicating effects of polymerization-depolymerization dynamics. In some regions of the channel, microtubules organized into asters (*Figure 1B*) as observed previously (*Verde et al., 1991*). A NUMA antibody was used to locate microtubule minus ends (*Mitchison et al., 2013*), and was found to localize to the aster core, confirming the polarity of the aster (*Gaglio et al., 1995*). Isolated asters were found to interact and coalesce (*Figure 1C*, *Video 1*). In other regions of the channel, microtubules formed networks of aster-like structures (*Figure 1D*), which were highly dynamic and exhibited large-scale motion that persisted for several tens of seconds (*Figure 1E*, *Video 2*). NUMA was found to localize to the interior of these structures, confirming their aster-like nature (*Figure 1F,G*).

To characterize these large-scale motions, we next imaged networks at lower magnification, obtaining a field of view spanning the entire channel width. The networks, which initially filled the

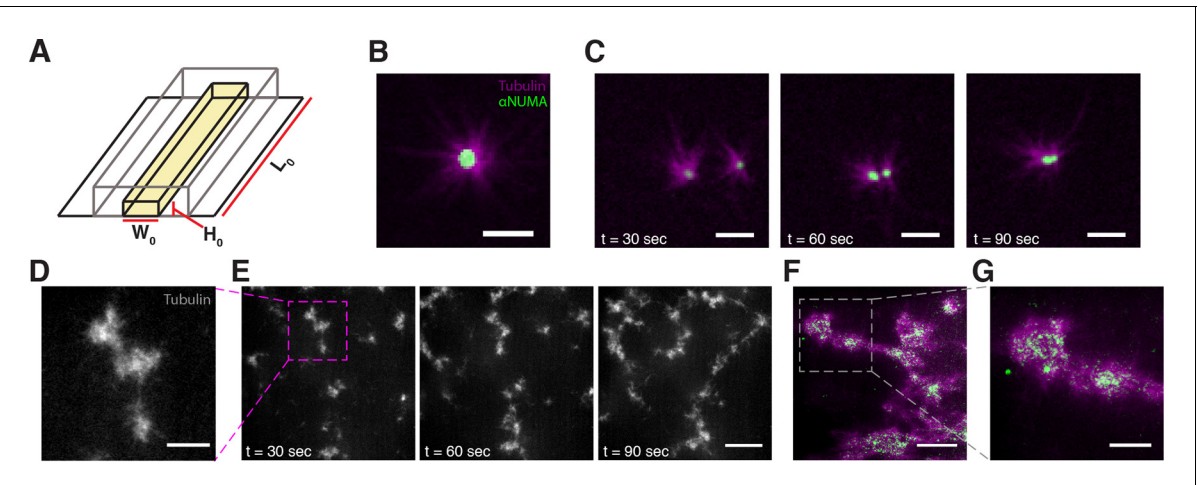

**Figure 1.** Stabilized microtubules form asters in *Xenopus* egg extracts. (**A**) Experiments were performed in thin rectangular channels of width $W_0$, height $H_0$, and length $L_0$. (**B**) In some regions of the channel, microtubules organize into asters, with minus ends localized in the aster core (Scale bar, 5 $\mu$m). (**C**) Isolated asters fuse together over minute timescales (Scale bar, 5 $\mu$m). (**D**) Aster-like structures form in other regions of the channel (Scale bar, 10 $\mu$m) (**E**) Aster-like structures show large scale movement on minute timescales. (Scale bar, 25 $\mu$m). (**F**) NUMA localizes to the network interior (Scale bar, 20 $\mu$m). (**G**) Closeup of aster-like structure showing NUMA localized on the interior (Scale bar, 10 $\mu$m).

entire channel (width $W_0$ = 1.4 mm), underwent a strong contraction, which was uniform along the length of the channel (*Figure 2A*, *Video 3*). The contractile behavior of these microtubule networks is highly reminiscent of the contractions of actin networks in these extracts (*Bendix et al., 2008*), but in our experiments actin filaments are not present due to the addition of 10 $\frac{\mu g}{mL}$ Cytochalasin D. We characterized the dynamics of microtubule network contractions by measuring the width, W(t), of the network as a function of time (*Figure 2B*). Occasionally, we observed networks tearing along their length (*Video 4*), yet these tears seemed to have little impact on the contraction dynamics far from the tearing site, arguing that the Poisson ratio of the network is ≈ 0. We then calculated the fraction contracted of the network:

$$\epsilon(t) = \frac{W_0 - W(t)}{W_0},$$  (1)

The time course of $\epsilon$(t) was found to be well fit by an exponential relaxation:

$$\epsilon(t) \simeq \epsilon_\infty \left[ 1 - e^{\frac{-(t-T_c)}{\tau}} \right],$$  (2)

where $\epsilon_\infty$ is the final fraction contracted, $\tau$ is the characteristic time of contraction, and $T_c$ is a lag time before contraction begins (*Figure 2B*, inset, *Figure 2—figure supplement 1*).

We next sought to investigate which processes determine the timescale of contraction and the extent that the network contracts. For this, we exploited the fact that different mechanisms predict different dependence of the timescale $\tau$ on the channel dimensions. For instance, in a viscoelastic Kelvin-Voight material driven to contract by a constant applied stress, $\tau = \eta/E$ depends solely on the viscosity $\eta$ and the Young's modulus E and is independent of the size of the channel (*Oswald, 2009*). In contrast, in a poroelastic material driven by a constant stress, $\tau \propto W_0^2$ (*Coussy, 2004*), where $W_0$ is the width of the channel. Thus, studying how $\tau$ varies with channel width provides a means to test the validity of these models.

We fabricated microfluidic channels of varying width, $W_0$ = 1.4, 0.9, 0.44, and 0.16 mm, all with height $H_0$ = 125 $\mu$m, loaded the channels with extracts supplemented with 2.5 $\mu$M Taxol, and imaged the networks at low magnification (*Figure 3A*, *Video 5*). Results for each channel width were averaged together to produce master curves of the width, W(t) (*Figure 3B*), and fraction contracted, $\epsilon$(t) (*Figure 3C*), of the networks in each channel. Visual inspection of the fraction-contracted curves, $\epsilon$(t), reveals that networks in smaller channels contract faster, but all reach a similar final

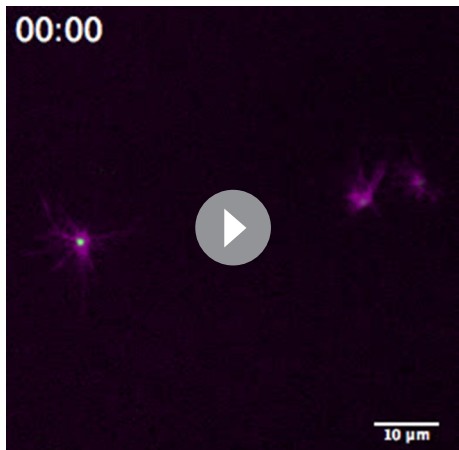

**Video 1.** Isolated asters undergo coalescence. Taxol stabilized microtubules in *Xenopus* oocyte extracts self-organize into asters that can then coalesce. The mageneta channel depicts microtubules while the green channel depicts NUMA localization, here used as a proxy for microtubule minus ends. Time is shown in minutes : seconds.

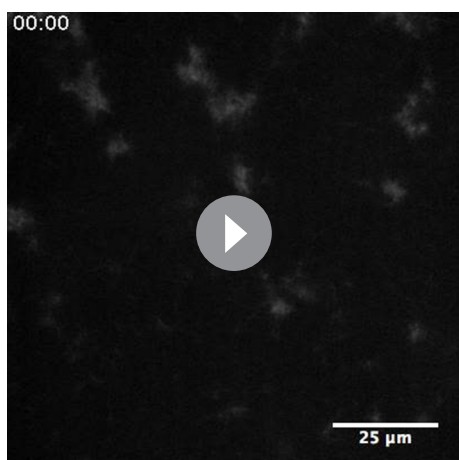

**Video 2.** Microtubules organize into dynamic aster-like structures. In other regions of the channel, microtubules organize into aster-like structures that exhibit large-scale movement on the minute timescale.
Time is shown in minutes : seconds.

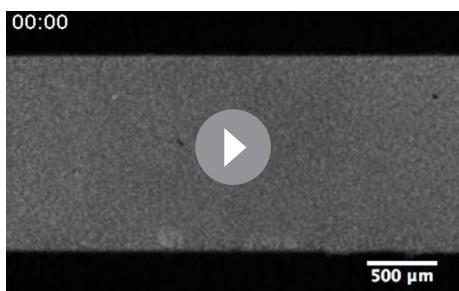

**Figure 2.** Stabilized microtubules form a contractile network in *Xenopus* egg extracts. (A) Low magnification imaging shows that microtubules form a contractile network (Scale bar, 500 $\mu m$). (B) The width of the microtubule network decreases with time (n = 6 experiments). (Inset) Representative plot of $\epsilon(t)$ (Blue line) and fit from (*Equation 2*) (Pink line), with $\epsilon_\infty = 0.81$, $\tau = 3.49$ min, $T_c = 1.06$ min.

The following figure supplement is available for figure 2:

**Figure supplement 1.** Plots of $\epsilon(t)$ from data in *Figure 1F* (Blue lines) along with fits from (*Equation 2*) (Pink lines).

fraction contracted (*Figure 3C*). To quantify these trends, we fit the $\epsilon(t)$ curves using (*Equation 2*) and extracted the characteristic time to contract, $\tau$, and the final fraction contracted, $\epsilon_\infty$, for each channel width. We find that the dependence of $\tau$ on channel width is inconsistent with the time of contraction resulting from either viscoelastic or poroelastic timescales, which would predict constant and quadratic scalings respectively (*Figure 3D*). We next explored the influence of channel height $H_0$ ($H_0$ = 75, 125, 150 $\mu$m, all with width $W_0$ = 1.4 mm) and found that $\tau$ does not significantly vary in these channels (*Figure 3E*).

In all cases, the networks contracted to a similar final fraction, $\epsilon_\infty$, of ≈ 0.77, irrespective of channel geometry (*Figure 3F*). Since the Taxol concentration was held constant, all experiments started with the same initial density of microtubules, regardless of the dimensions of the channel. Thus, all networks contracted to the same final density. By using fluorescence intensity as a proxy for tubulin concentration (see Materials and methods), we estimate the final concentration of tubulin in the network to be $\rho_0$ ≈ 30 $\mu$M. Remarkably, this is comparable to the concentration of microtubules in reconstituted meiotic spindles in *Xenopus* extracts (*Needleman et al., 2010*), which is ≈ 60 $\mu$M. As neither the simple viscoelastic nor poroelastic models are consistent with these results, we sought to construct an alternative model of the contraction process. Since Taxol stabilizes microtubules in these experiments, the density of microtubules $\rho$ is conserved throughout the contraction process, implying

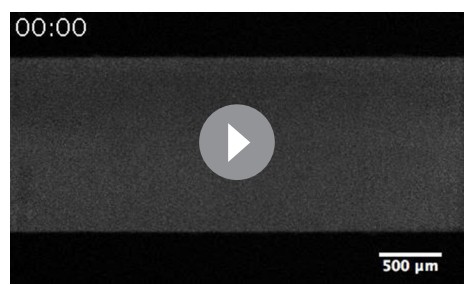

**Video 3.** Microtubule networks undergo a spontaneous bulk contraction. Low magnification imaging of the channels reveals that microtubules organize into a macroscopic network that spontaneously contracts on the millimeter length scale. Time is shown in minutes : seconds.

**Video 4.** Microtubule networks can undergo tearing. During contraction, tears can develop in the microtubule network, causing the network to break. Time is shown in minutes : seconds.

$$\partial_t \rho = -\vec{\nabla} \cdot (\rho \vec{v}), \tag{3}$$

where $\vec{v}$ is the local velocity of the microtubule network. The velocity $\vec{v}$ is set by force balance. If the relevant timescales are long enough that the microtubule network can be considered to be purely viscous, and if the network's motion results in drag, then the equation for force balance is

$$\eta \nabla^2 \vec{v} - \gamma \vec{v} = \vec{\nabla} \cdot \sigma, \tag{4}$$

where $\eta$ and $\gamma$ are the viscosity and drag coefficients, respectively, and $\sigma$ is an active stress caused by motor proteins which drive the contraction of the microtubule network. The observation that the timescale of contraction, $\tau$, is independent of channel height (*Figure 3E*) shows that the drag does not significantly vary with channel height, and thus could arise from weak interactions between the microtubule network and the device wall.

We obtain an expression for the active stress, $\sigma$, by considering the microscopic behaviors of microtubules and motor proteins. As the contracting networks consist of microtubule asters (*Figure 1D, E*), and microtubule asters in meiotic extracts are thought to assemble by the dynein-induced clustering of microtubule minus ends (*Verde et al., 1991*), we hypothesize that the contraction process is also driven by dynein pulling microtubule minus ends towards each other (*Figure 4A*).

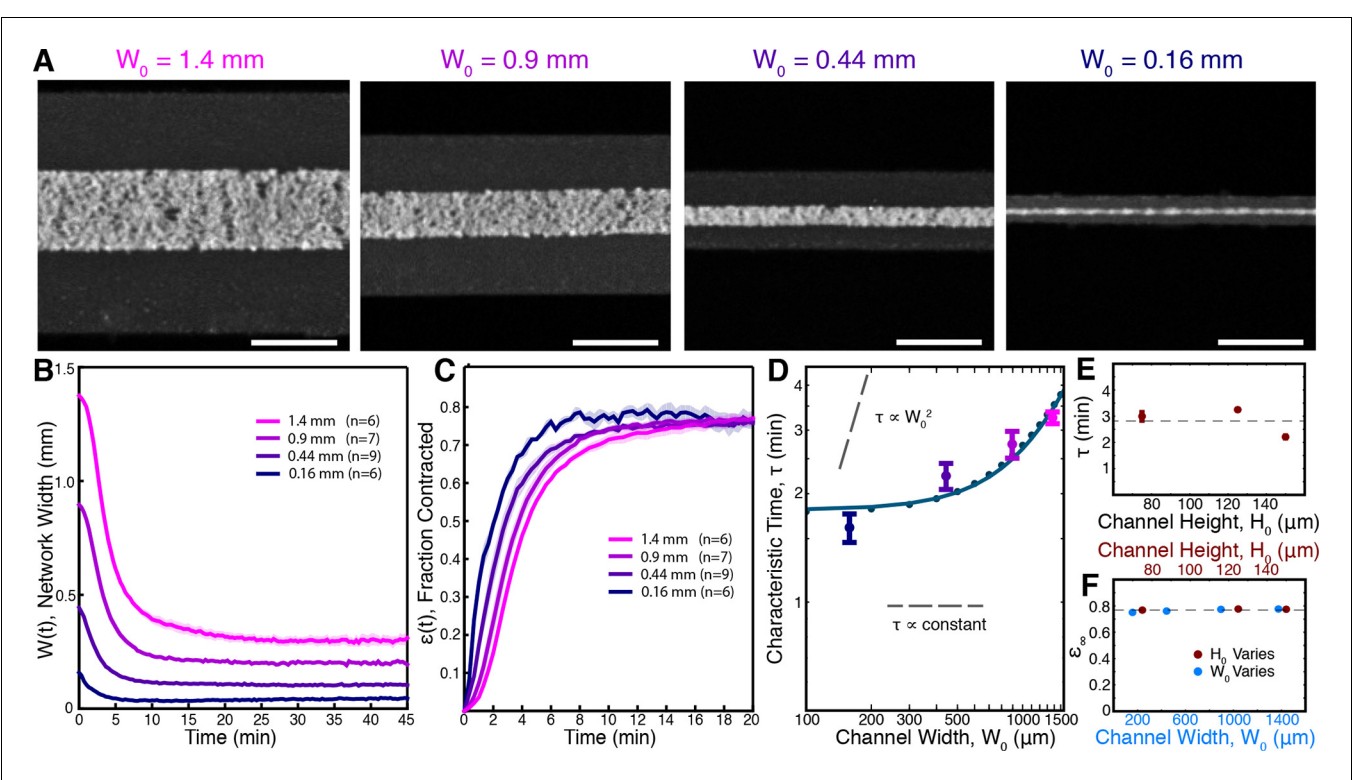

**Figure 3.** Contraction dynamics in channels of different width provide a means to test potential contraction mechanisms. (A) Microtubules form contractile networks in channels with various widths (Scale bar, 500 $\mu$m, t=10 min). (B) Width of the networks as a function of time in channels with various widths. (C) Fraction contracted as a function of time, $\epsilon(t)$, calculated from the data in B. The networks all contract to a similar final fraction, while the timescale of contraction differs. (D) The scaling of the characteristic time, $\tau$, with channel width does not vary as $W_0^2$, as would result for a poroelastic timescale, and is not a constant, independent of width, as would result from a viscoelastic timescale. The scaling is well described by an active fluid model (green line analytic scaling, fit to (*Equation 6*); green dots numerical solution). (E) The characteristic time, $\tau$, is found to be independent of channel height. The dashed line is the mean value of $\tau$. (F) $\epsilon_\infty$ is constant for all channel widths and heights, indicating that the network contracts to a constant final density. The dashed line is the mean value of $\epsilon_\infty$. All panels display mean $\pm$ s.e.m.

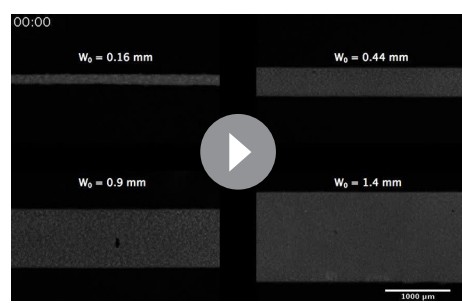

**Video 5.** Network contraction in channels of varying width. Devices were fabricated with different widths. Each video panel depicts a representative experiment using channels of the given width. Time is shown in minutes : seconds.

In an orientationally disordered suspension of microtubules, we expect dynein mediated collection of microtubule minus ends to drive a contractile stress which is proportional to the number of motor molecules $m$ and the local density of microtubules $\rho$, (see Appendix).

As only a finite number of microtubules can fit near the core of an aster, steric collisions will counteract the contractile stress at high densities (**Figure 4B**).

Since most motion in the suspension is motor driven, thermal collisions can be ignored, and the extensile stress driven by steric interactions will be be proportional to the number of motor molecules $m$ and quadratic in the local density of microtubules $\rho$ (see Appendix).

Taken together, these two effects lead to the active stress

$$\sigma = s\rho(\rho - \rho_0)\mathbb{I}, \tag{5}$$

where $s$ is the strength of the active stress, $\rho_0$ is the final density at which the effects of dynein mediated clustering and steric repulsion between microtubules balance, and $\mathbb{I}$ is a unit tensor (see Appendix).

Importantly, since the contractile and extensile parts of the active stress both depend linearly on the number of motor molecules, the prefered density $\rho_0$ that the suspension will reach after contraction depends only on the interaction geometry between microtubules and motors and not on the actual number of active motors. Only the strength $s$ of the active stress will be affected if the number of active motors could be changed.

Taken together, **Equations (3,4,5)** constitute an active fluid theory of microtubule network contraction by minus end clustering. We note that this theory could be reformulated, essentially without change, as the clustering of aster cores, again driven by dynein mediated clustering of minus ends. Isotropy of interactions remains a fundamental assumption.

We first investigated if this active fluid theory can explain the dependence of the timescale of contraction on sample geometry. An analysis of the equations of motion, **Equations (3,4,5)**, near equilibrium predicts that the timescale of contractions obeys

$$\tau(W_0) = \alpha\frac{\eta}{s\rho_0^2} + \beta\frac{\gamma}{s\rho_0^2}W_0^2, \tag{6}$$

where $\alpha = 2.2 \pm 0.05$ and $\beta = 0.085 \pm 0.006$ are dimensionless constants, which we determined numerically (see Appendix). This predicted scaling is both consistent with the experimental data and simulations of the full theory (**Figure 3D**). Fitting the scaling relationship to the data allows combinations of the parameters to be determined, giving $\eta/(s\rho_0^2) = 0.82 \pm 0.20$ min and $\gamma/(s\rho_0^2) = 1.0 \times 10^{-5} \pm 0.7 \times 10^{-5}$ min/$(\mu m^2)$ (mean $\pm$ standard error). Combining this measurement with an estimate for the network viscosity taken from measurements in spindles of $\eta \approx 2 \times 10^2 Pa\cdot s$ (**Shimamoto et al., 2011**), we can estimate the dynein-generated active stress to be $s\rho_0^2 \approx 4 Pa$ which is consistant with having $\approx 0.4$ dynein per microtubule minus end each exerting an average force of 1 pN (**Nicholas et al., 2015**).

To further explore the validity of the active fluid theory of contraction by microtubule minus end clustering, we explored other testable predictions of the theory. This theory predicts that: (i) the preferred density of the network $\rho_0$ is constant and does not depend on the initial conditions. This is consistent with the constant $\epsilon_\infty$ measured experimentally (**Figure 3F**); (ii) since contractions are driven by stress gradients (**Equation 4**) and stress depends on microtubule density (**Equation 5**) the density discontinuity at the edge of the network should produce large-stress gradients, leading to an inhomogeneous density profile in the network during contraction; (iii) the magnitude of the active stress, $s$, is proportional to the number of active motors, but the final density of the network, $\rho_0$, is

independent of the number of molecular motors (see Appendix). Thus, reducing the number of motors should lead to slower contractions, but still yield the same final density.

We first examined prediction (ii), that the stress discontinuity at the edge of the network should lead to a material buildup in the film. To test this, we averaged the fluorescence intensity along the length of the channel (see Materials and Methods) and found that the microtubule density does indeed increase at the network's edge during contraction (*Figure 5A*). We next explored if the inhomogeneous density profile could be quantitatively explained by our active fluid theory. We numerically solved *Equations (3,4,5)* and used least squares fitting to determine the simulation parameters which most closely matched the experimentally measured profiles (*Figure 5B*), yielding $\eta/(s\rho_0^2)$ = 0.82±0.03 min, $\gamma/(s\rho_0^2)$ = 6.1±0.1×$10^{-6}$ min/($\mu m^2$), and $\rho_{initial}/\rho_0$ = 0.32 ± 0.01 (mean ± s.e.m., n=4 experiments). Within error, these values are the same as those determined from the dependence of the timescale of contraction on channel width (*Figure 3D*). The simulated profiles closely match the experimental ones for most of the contraction (*Figure 5B*), but at late times the simulated inhomogeneities dissipate in contrast to the experiments (*Figure 5—figure supplement 1*). This might be caused by a long-term aging of the network that is not incorporated into our simple model. To confirm that the density buildup was due to an increased velocity near the network's edge, we measured the velocity throughout the network using Particle Image Velocimetry (PIV, see Materials and Methods) (*Figure 5C*) and found that the velocities increase superlinearly with distance from the network's center, as predicted (*Figure 5D*).

Finally, we sought to determine the molecular basis of the contraction process, and check prediction (iii), that the number of motors driving the contraction affects the rate of contraction, but not

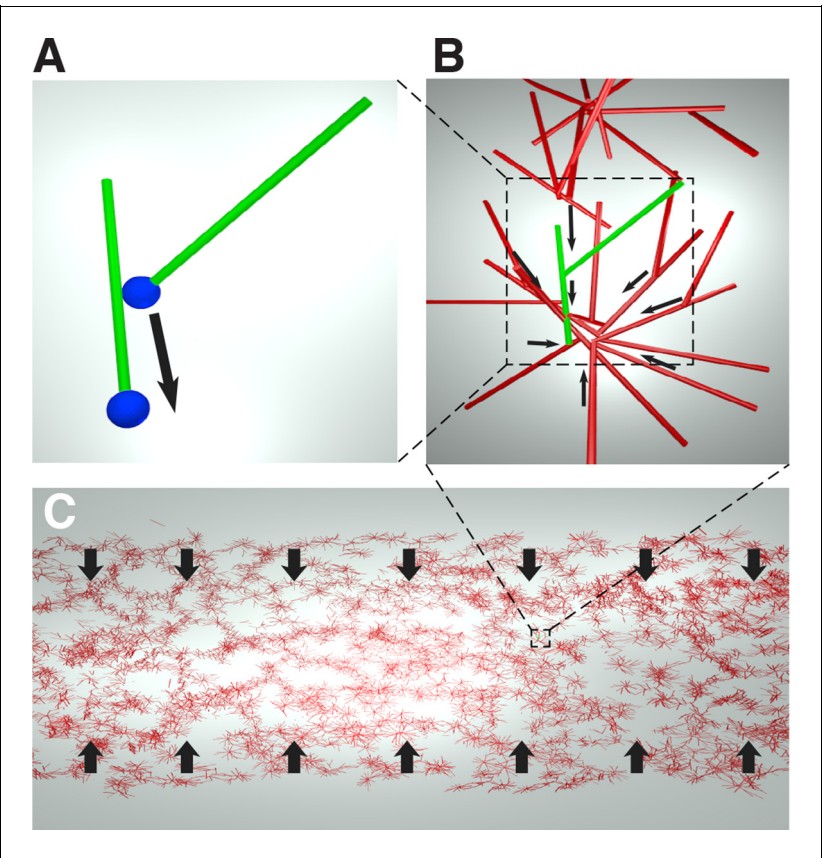

**Figure 4.** Cartoon of the microscopic model underlying the active fluid theory of network contractions by minus end clustering. (A) Microtubule sliding by dynein drives microtubule minus ends together. (B) Minus end clustering leads to the formation of aster-like structures. Due to steric interactions between microtubules, there is an upper limit to the local microtubule density. (C) The microtubule network is composed of interacting asters. Motor activity driving aster cores together leads to bulk contraction of the network.

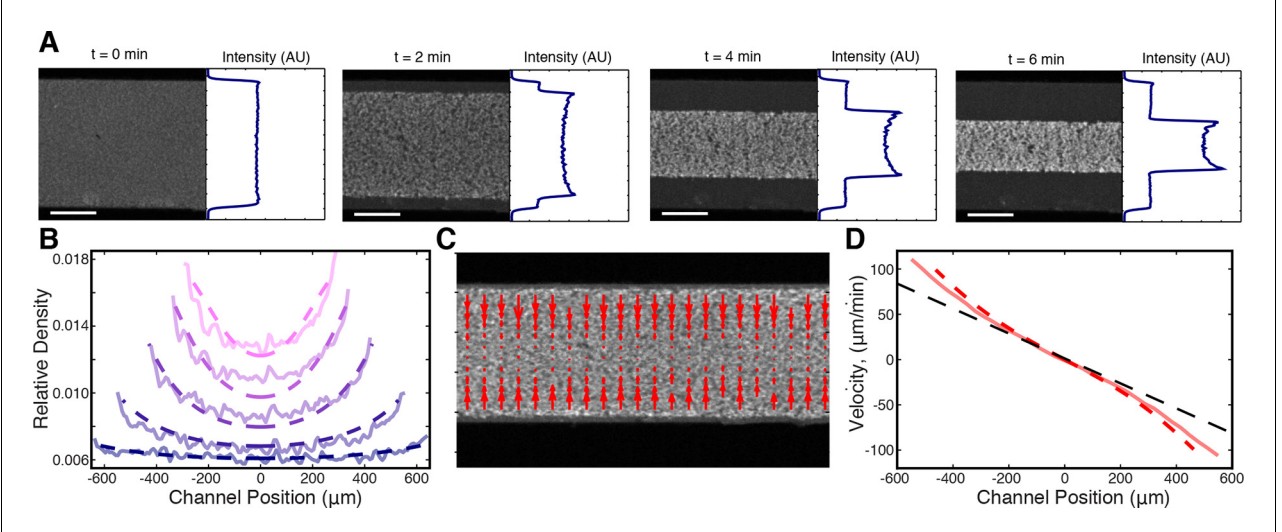

**Figure 5.** Microtubule density increases at the network's edges during contraction. (A) Time series of contraction showing intensity averaged along the length of the channel. The average intensity peaks at the network's edges due to increased local microtubule density. (Scale bars, 500 $\mu$m) (B) Comparison of measured density profiles (solid lines) with density profiles from simulation (dashed lines). Data are plotted at 1 min intervals starting at t = 40 s. (C) Representative frame from PIV showing the network's local velocity component along the network's width. (D) Comparison between measured (solid red line) and simulated (dashed red line) velocity along the width of the channel at t = 80 s. The measured and simulated velocities increase superlinearly with distance from the center of the network, as can be seen by comparison to a linear velocity profile (dashed black line).

The following figure supplement is available for figure 5:

**Figure supplement 1.** Comparison between measured (solid lines) and simulated (dashed lines) density profiles.

the final density the network contracts to. Aster assembly is dynein-dependent in *Xenopus* egg extracts (*Gaglio et al.,1995*; *Verde et al., 1991*), and dynein (*Heald et al., 1996*) and Kinesin-5 (*Sawin et al., 1992*) are two of the most dominant motors in spindle assembly in this system. We inhibited these motors to test their involvement in the contraction process. Extracts supplemented with STLC for Kinesin-5 inhibition or p150-CC1 for dynein inhibition were loaded into channels with a width, $W_0$, of 0.9 mm and imaged at low magnification. Inhibiting Kinesin-5 had little effect on the contraction process (*Figure 6—figure supplement 1*). In contrast, inhibiting dynein caused a dose-dependent slowdown of the contraction (*Figure 6A*). In spindle assembly, inhibiting Kinesin-5 suppresses the morphological changes caused by dynein inhibition (*Mitchison et al., 2005*). We, therefore, tested how simultaneously inhibiting both motors influences the contraction process, but found that the effects of dynein inhibition were not rescued by the simultaneous inhibition of Kinesin-5 (*Figure 6—figure supplement 1*), suggesting that in this context, Kinesin-5 is not generating a counteracting extensile stress. This further suggests the possibility that in the spindle, the role of Kinesin-5 may be in orienting, polarity sorting, and sliding microtubules as opposed to active stress generation. Curves of $\epsilon$(t) were fit using *Equation (2)* to extract the final fraction contracted, $\epsilon_\infty$, and the characteristic time of contraction, $\tau$. By varying the concentration of p150-CC1, the characteristic time, $\tau$, could be tuned over a wide range from $\approx$ 3 min to $\approx$ 75 min (*Figure 6B*). Fitting a sigmoid function to the $\tau$ vs. p150-CC1 concentration curve yields an EC50 value of 0.22 $\pm$. 02 $\mu$M (mean $\pm$ standard error), similar to the value of $\approx$ 0.3 $\mu$M reported for the effect of p150-CC1 on spindle length in *Xenopus* extracts (*Gaetz and Kapoor, 2004*), which is consistent with active stress generated by dynein being required for pole focusing. Despite this large change in the contraction timescale, we found no apparent differences in $\epsilon_\infty$ (*Figure 6C*). Thus, the microtubule networks contract to approximately the same final density irrespective of the concentration of p150-CC1. The observation that inhibiting dynein affects the timescale of contraction but not the final density to which the network contracts is consistent with the predictions of our model. We note that even at the highest p150-CC1 concentrations used, the network still undergoes a bulk contraction. This could possibly be due to incomplete inhibition of dynein by p150-CC1, or by another motor protein present in the

extract that also contributes to the contraction process. As the characteristic time, $\tau \propto \frac{1}{s}$, by comparing the characteristic times in the uninhibited and 2 $\mu$M p150-CC1 cases, we can estimate that the strength of the active stress, $s$, in the 2 $\mu$M p150-CC1 condition is only $\approx$ 4% of the strength of the active stress in the uninhibited case, arguing that even if another motor is involved in the contraction, dynein contributes $\approx$ 96% of the active stress.

## Discussion

Here, we have shown that networks of stabilized microtubules in *Xenopus* egg extracts undergo a bulk contraction. By systematically varying the width of the microfluidic channel in which the network forms, we demonstrated that the timescale of contraction is not a poroelastic or viscoelastic timescale. A simple active fluid model of network contraction by dynein-driven clustering of microtubule minus ends correctly predicts the dependence of the contraction timescale on channel width, the nonuniform density profile in the network during contraction, and that inhibiting dynein affects the timescale of contraction but not the final density that the network contracts to. Parameters of this model can be measured by the scaling of the contraction timescale with channel width and by a detailed analysis of the inhomogeneities in the network that develop during contraction. Both methods give similar values.

Our results demonstrate that the behaviors of a complex biological system can be quantitatively described by a simple active matter continuum theory. These active matter theories aim to describe the behavior of cytoskeletal systems at large-length scales and long-timescales by effectively averaging all of the molecular complexity into a small set of coarse-grained parameters. Previously, these theories have been predominately applied to describe biological systems near non-equilibrium steady states (*Prost et al., 2015*; *Brugués et al., 2014*). In the present work, we augment previous theories with a nonlinear active stress term derived from microscopic considerations to capture the far from steady state dynamics of the contraction process. This approach allows us to quantitatively explain our experimental results using a theory with only four parameters, while a complete microscopic model would require understanding the behavior of the thousands of different proteins present in *Xenopus* egg extracts. Furthermore, the considerations of the model are general, and it will be interesting to consider whether the end clustering mechanism proposed here could contribute to contraction in actin networks as well.

In our model, the active stress which drives network contraction results from the motor-induced clustering of microtubule minus ends, the same process thought to be responsible for aster formation and spindle pole focusing (*Gaglio et al., 1995*; *Mountain et al., 1999*; *Verde et al., 1991*, *Elting et al., 2014*; *Heald et al., 1996*; *Burbank et al., 2007*; *Khodjakov et al., 2003*; *Goshima et al., 2005*). Our results, and previous data (*Verde et al., 1991*; *Heald et al., 1996*; *Burbank et al., 2007*), are consistent with minus end clustering in *Xenopus* egg extracts primarily arising from the activity of dynein. The ability of dynein to cluster microtubule minus ends could result from dynein being able to accumulate on the minus end of one microtubule, while simultaneously walking towards the minus end of another (*Hyman and Karsenti, 1996*; *McKenney et al. 2014*; *Figure 4A*). There is indication that such behaviors may indeed occur in spindles (*Elting et al., 2014*), and pursuing a better understanding of those processes is an exciting future direction that will help to clarify the function of dynein in spindles.

The observation that microtubule networks contract in *Xenopus* egg extracts suggests that motor-induced stresses in spindles are net contractile and not extensile as previously assumed (*Brugués and Needleman, 2014*). The contribution of dynein to spindle pole focusing may ultimately be due to these contractile stresses. The presence of contractile stresses from dynein might also explain both the observation that the fusion of spindles is dynein-dependent (*Gatlin et al., 2009*), and the apparently greater cohesion between microtubules at spindle poles, (where dynein is localized [*Gatlin et al., 2010*]). It is unclear what processes set the density of microtubules in the spindle, and the finding that the active stress generated from minus end clustering saturates at a preferred microtubule density could play an important role.

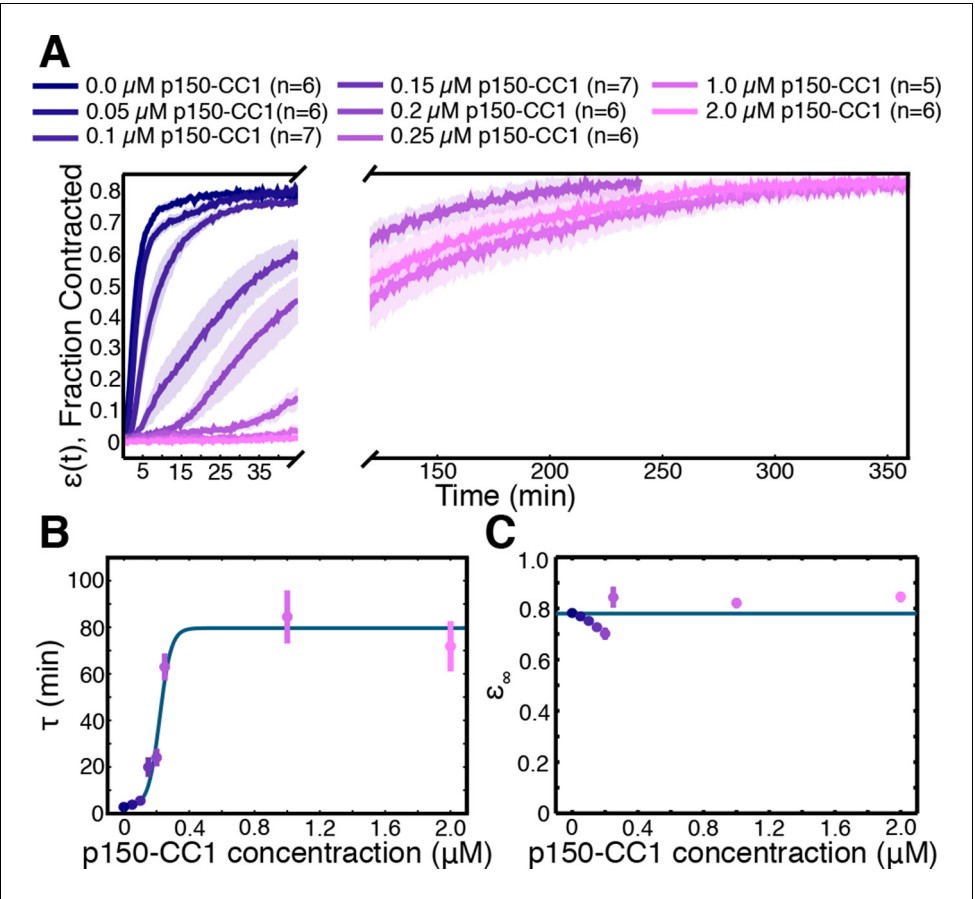

**Figure 6.** Network contraction is a dynein-dependent process. (**A**) Fraction contracted as a function of time, $\epsilon(t)$, when dynein is inhibited using p150-CC1. (**B**) The characteristic time of contraction, $\tau$, increases with increasing p150-CC1 concentration. Solid green line indicates fit of sigmoid function. (**C**) $\epsilon_\infty$ has no apparent variation with p150-CC1 concentration. Solid green line indicates the mean value of $\epsilon_\infty$. All panels display mean $\pm$ s.e.m.

The following figure supplements are available for figure 6:

**Figure supplement 1.** Inhibition of Kinesin-5 has little effect on the contraction process.

**Figure supplement 2.** Plots of $\epsilon(t)$ from experiments with 2 $\mu$M p150-CC1 (blue lines) along with fits from *Equation (2)* (pink lines).

## Materials and methods

### Preparation of *Xenopus* extracts

CSF-arrested extracts were prepared from *Xenopus llaevis* oocytes as previously described (*Hannak and Heald, 2006*). Crude extracts were sequentially filtered through 2.0, 1.2, and 0.2 micron filters, frozen in liquid nitrogen, and stored at −80°C until use.

### Preparation of microfluidic devices

Channel negatives were designed using AutoCAD 360 (Autodesk) and Silhouette Studio (Silhouette America) software, cut from 125-micron-thick tape (3M Scotchcal, St. Paul, MN) using a Silhouette Cameo die cutter, and a master was made by adhering channel negatives to the bottom of a petri dish. PDMS (Sylgard 184, Dow Corning, Midland, MI; 10:1 mixing ratio) was cast onto the masters and cured overnight at 60°C. Devices and coverslips were each corona treated with air plasma for 1 min before bonding. Channels containing a degassed solution of 5 mg/mL BSA (J.T. Baker, Center

Valley, PA) supplemented with 2.5% w/w Pluronic F127 (Sigma, St. Louis, MO) were incubated over-night at 12°C. Unless stated otherwise, the microfluidic devices had a length of 18 mm, a height of 0.125 mm, and a width of 1.4 mm.

## Protein purification

GST-tagged p150-CC1 plasmid was a gift from Thomas Surrey (*Uteng et al., 2008*). GST-p150-CC1 was expressed in *E. coli* BL21 (DE3)-T1$^R$(Sigma) for 4 hr at 37°C. The culture was shifted to 18°C for 1 hr before adding 0.2 mM IPTG and the culture was grown overnight at 18°C. Cells were centri-fuged, resuspended in PBS supplemented with Halt Protease Inhibitor Cocktail (Thermo Scientific, Rockford, IL) and benzonase (Novagen, San Diego, CA) before lysis by sonication. GST-p150-CC1 was purified from clarified lysate using a GSTrap column FF (G.E. Healthcare, Sweden) as per the manufacturer's instructions. GST-p150-CC1 was dialyzed overnight into 20 mM Tris-HCl, 150 mM KCl, and 1 mM DTT. The GST tag was cleaved using Prescission Protease (overnight incuba-tion at 4°C). After removing free GST and Prescission Protease using a GSTrap FF column, p150-CC1 was concentrated, frozen in liquid nitrogen, and stored at -80°C until use.

## Bulk contraction assay

20 $\mu$L aliquots of filtered extract were supplemented with ~1 $\mu$M Alexa-647 labeled tubulin and 2.5 $\mu$M Taxol before being loaded into channels. For dynein inhibition experiments, 1 $\mu$L of either p150-CC1 or buffer alone was added to the extract immediately before Taxol addition. For Kinesin-5 inhibition experiments, 100 $\mu$M STLC (Sigma Aldrich) was added concurrently with Taxol. Channels were sealed with vacuum grease and imaged using a spinning disk confocal microscope (Nikon Ti2000, Yokugawa CSU-X1), an EMCCD camera (Hamamatsu), and a 2x objective using Metamorph acquisition software (Molecular Devices). t=0 is defined when the imaging begins, ≈ 1 min after Taxol addition to the extract. After a brief lag time, the microtubule networks spontaneously begin con-traction. Images were analyzed using ImageJ and custom build MATLAB and Python software (avail-able at https://github.com/peterjfoster/eLife). Parameters were fit to contraction data using timepoints where $\epsilon(t) > 0.1$.

## Final density estimation

The final density was estimated using contraction experiments with 2.5 $\mu$M Taxol in 0.9 mm chan-nels. For each experiment, the frame closest to t = $\tau + T_c$ was isolated, where $\tau$ and $T_c$ are the time-scale of contraction and the offset time respectively, obtained from fits of the time course of contraction to *Equation 2* of the main text. After correcting for the camera offset and inhomoge-neous laser illumination, the average fluorescence intensity of the network, $\rho_N$ and the average fluo-rescence intensity in the channel outside the network, $\rho_M$ were calculated. The fluorescence intensity in the channel but outside the network comes from monomeric fluorescently labeled tubulin and was assumed to be constant throughout the channel. The fractional concentration was then estimated as $\rho(\tau + T_c) = \frac{\rho_N - \rho_M}{\rho_N + \rho_M}$. Using this measurement along with the fit curves for $\epsilon(t)$ and under the assumption that the network contracts in the z direction such that $\epsilon(t)$ in the z direction is the same as along the width, the inferred fractional concentration at t = $\infty$ was calculated as

$$\rho(t=\infty) = \frac{\rho(\tau + T_c)}{(1 - \epsilon_\infty)^2} \left(1 - \epsilon_\infty(1 - e^{-1})\right)^2$$

Assuming the fluorescently labeled tubulin incorporates into microtubules at the same rate as endogenous tubulin, we can multiply the derived fractional density $\rho$(t = $\infty$) by the tubulin concentra-tion in extract, ≈18 $\mu$M (*Parsons and Salmon, 1997*) to arrive a final network tubulin concentration of ≈30 $\mu$M.

## Density profile measurements

Images from contraction experiments were corrected for the camera offset and inhomogeneous laser illumination before being thresholded in order to segment the microtubule network from back-ground fluorescence. Rotations of the channel relative to the CCD were detected by fitting linear equations to edges of the microtubule network. If the average of the slopes from the top and bot-tom of the network was greater than 1/(the number of pixels in the length of the image), a rotated,

interpolated frame was constructed where pixels were assigned based on the intensity of the pixel in the original frame weighted by their area fraction in the interpolated pixel. Frames were averaged along the length of the channel before background signal subtraction. For density profiles compared with simulations, the edge peaks of the density profile were identified and pixels between the two peaks were retained. Profiles were normalized such that the integral of the profile was set to 1.

### Particle imaging velocimetry

Particle Imaging Velocimetry was performed using PIVLab software (*Thielicke and Stamhuis, 2014*) using the FFT window deformation algorithm with a 16-pixel interrogation area and 8 pixel step for the first pass and an 8 pixel interrogation area with a 4-pixel step for the second pass. After PIV was performed, intensity images were thresholded to segment the microtubule network from the background, and only velocity vectors within the microtubule network that were > 8 pixels from the network's edges were retained.

## Acknowledgements

The authors would like to thank Bryan Hassell for assistance fabricating the microfluidic devices, Thomas Surrey for the generous gift of the GST-p150-CC1 plasmid, and Tim Mitchison for the gift of labeled NUMA antibody. SF acknowledges support by Human Frontiers Science Program. This work was supported by National Science Foundation Grants PHY-0847188, PHY-1305254, and DMR-0820484 to DJN and Grant DMR-1420073 to MJS, and National Institutes of Health Grant 1R01GM104976-01 to MJS.

## Additional information

### Funding

| Funder | Grant reference number | Author |
| --- | --- | --- |
| National Science Foundation | PHY-0847188 | Daniel J Needleman |
| National Science Foundation | PHY-1305254 | Daniel J Needleman |
| National Science Foundation | DMR-0820484 | Daniel J Needleman |
| National Science Foundation | DMR-1420073 | Michael J Shelley |
| National Institutes of Health | 1R01GM104976-01 | Michael J Shelley |
| Human Frontier Science Program | HFSP Postdoctoral Fellowship | Sebastian Fürthauer |

The funders had no role in study design, data collection and interpretation, or the decision to submit the work for publication.

### Author contributions

PJF, Conception and design, Acquisition of data, Analysis and interpretation of data, Drafting or revising the article; SF, MJS, DJN, Conception and design, Analysis and interpretation of data, Drafting or revising the article

### Author ORCIDs

Peter J Foster, http://orcid.org/0000-0003-1818-5886

### Ethics

Animal experimentation: All animals were handled according to approved institutional animal care and use committee (IACUC) protocols (#28-18) of Harvard University.

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

## Appendix

<div style="background-color:#eaf4fb; padding:1em;">

# Derivation of active fluid theory

We introduce a theoretical description of a confined active microtubule-motor gel immersed in a Newtonian fluid. We obtain generic equations of motion for this system closely following the logic outlined in (*Doi and Onuki, 1992*; *Joanny et al., 2007*). This generic description is augmented by including a density-dependent active stress, which is derived from a minimal microscopic description of microtubule-dynein interactions. Here, we present the equations of motion for the one dimensional system.

## Generic theory for an immersed active gel

We begin by stating the conservation laws an active gel permeated by a Newtonian fluid obeys. The system shall be incompressible such that the total density $\rho_{tot} = \rho + \rho_f$ is a constant. Here $\rho$ and $\rho_f$ are the densities of gel and fluid, respectively. The gel density $\rho$ obeys the continuity equation,

$$\partial_t \rho = -\partial_x(\rho v), \tag{7}$$

where $v$ is the velocity of the gel. Similarly the fluid permeating the gel obeys,

$$\partial_t \rho_f = -\partial_x(\rho_f v_f), \tag{8}$$

where $v_f$ and $\rho_f$ are the fluid density and velocity fields, respectively. Since the overall system is incompressible $\partial_x(\rho v + \rho_f v_f) = 0$. Force balance in the gel requires

$$\partial_x \sigma^{gel} = \bar{\gamma} v + \lambda(v - v_f), \tag{9}$$

where the gel stress $\sigma^{gel} = \eta \partial_x v - \sigma + (\rho/\rho_{tot})P$ consists of a viscous stress $\eta \partial_x v$, an active stress $\sigma$, and a hydrostatic pressure $P$. The friction coefficient $\bar{\gamma}$ quantifies the momentum transfer between the gel and its confinement and $\lambda$ quantifies the momentum transfer between and the gel and the fluid. The momentum continuity equation of the permeating fluid is

$$0 = \eta_f \partial_x^2 v_f - \partial_x\left[(\rho_f/\rho_{tot})P\right] + \lambda(v - v_f), \tag{10}$$

where $\eta_f$ is the fluid viscosity. In our experiments, changing the height of the chamber does not appreciably change the timescale $\tau$ of the observed contractions, see *Figure 3F*. Furthermore, there is little observed motion of the extract surrounding the film. Presumably $v_f \ll v$ since the system is relatively dilute, i.e. $\rho \ll \rho_f$, and the length-scale $\sqrt{\eta_f/\lambda}$ is large compared to the chamber height. We thus simplify *Equation (9)* to

$$\eta \partial_x^2 v - \gamma v = \partial_x \sigma, \tag{11}$$

where $\gamma = \lambda + \bar{\gamma}$. *Equation (11)* is the force balance equation we henceforth use for the gel to quantitatively capture the experimental dynamics. Note that $\rho \ll \rho_f$ also allowed us to neglect the hydrostatic pressure in the gel. We complement *Equation (11)* by the stress boundary condition at the edges of the film at $x = \pm W(t)/2$

</div>

$$[\eta \partial_x v - \sigma]_{x=\frac{\pm W(t)}{2}} = 0. \tag{12}$$

The width of the film obeys

$$\partial_t W(t) = v(W(t)/2) - v(-W(t)/2). \tag{13}$$

## Active stresses from dynein-mediated microtubule interactions

We next seek to obtain an expression for the active stress by coarse-graining a microscopic model of interactions between dynein molecular motors and microtubules. Here, we assume that dynein builds up near microtubule minus ends as previously suggested (*Elting et al., 2014*; *Surrey et al., 2001*), and hence forces are exchanged between microtubules through the microtubules' minus ends. We introduce the positions of microtubule minus ends $x_i$, such that the film density can be written as

$$\rho(x) = \sum_i \delta(x - x_i). \tag{14}$$

The force exerted by the $i$-th on the $j$-th filament is $F_{ij}$, with $F_{ij} = -F_{ji}$ as required by momentum conservation. The active stress $\sigma$ generated in this context is defined by the force balance equation

$$\partial_x \sigma = \sum_i \delta(x - x_i) \sum_j F_{ij}, \tag{15}$$

up to an arbitrary constant of integration. Note that averaging *Equation (15)* over an appropriate mesoscopic volume yields the well-known Kirkwood formula. To model microtubule-dynein interactions, we propose that $F_{ij} = A_{ij} + R_{ij}$, where $A_{ij}$ is a dynein-mediated attractive force between minus ends, and $R_{ij}$ is a repulsive force from steric interactions between nearby filaments (*Figure 4*). Generically, $A_{ij}$ and $R_{ij}$ depend on the relative positions and orientations of microtubules $i$ and $j$. Since we are concerned with a disordered assembly of microtubules in which all orientations occur with the same likelihood it is sufficient for our purposes to only think of microtubule positions, and orientation effects average out. The average attractive force $A_{ij}$ that motors bound to the minus end of filament $i$ exert on filament $j$ can be expressed locally as the product

$$A_{ij} = (P_{ij} + P_{ji})a_{ij}, \tag{16}$$

where $P_{ij}$ is the probability that a motor connects the minus end of filament $i$ to filament $j$ and $a_{ij}$ is the force which the motor exerts if a connection is made. Since each dynein can link at most two filaments,

$$P_{ij} = m \frac{1 - \Theta(|x_i - x_j| - \Gamma)}{\sum_{k \neq i} \left(1 - \Theta(|x_i - x_k| - \Gamma)\right)}, \tag{17}$$

where $m$ is the fraction of filaments that carry an active motor at their minus end and $\Gamma$ is a typical interaction distance. Here, $\Theta(x)$ denotes the Heaviside function which is equal to one for positive $x$ and zero otherwise. If $a_{ij}$ is an odd function of the separation vector $x_i - x_j$, it can be expressed by the series $a_{ij} = \sum_{n \geq 1} A_n (x_i - x_j)^{2n-1}$. Using *Equation (14)*, the force density field generated by motor contractions becomes to lowest order in $\Gamma$,

$$\sum_i \delta\,(x - x_i)\,\sum_j A_{ij} = m A_1 \frac{2\Gamma^2}{3}\partial_x\rho \;+\; \mathcal{O}\!\left(\Gamma^4\right),\tag{18}$$

which corresponds to an active stress contribution $(2/3)m A_1 \Gamma^2 \rho$.

We next discuss the average steric force that filament $i$ exerts on filament $j$. Given the force $r_{ij}$ of a collision event we find

$$R_{ij} = m(1 - \Theta(\,|\,x_i \,-\, x_j\,|\, - \Gamma))r_{ij}.\tag{19}$$

**Equation (19)** is linear in the motor density $m$, since only filaments that are being actively moved will sterically displace their neighbors. Note that here we chose the typical interaction distance $\Gamma$ to be the same in **Equation (19)** and **Equation (16)** for simplicity. If $r_{ij} = \sum_{n\geq 1} R_n(x_i - x_j)^{2n-1}$ is an odd function of the displacement between the microtubule ends $i$ and $j$ the force density field generated by steric interactions is

$$\sum_i \delta\,(x - x_i)\,\sum_j R_{ij} = m R_1 \frac{2\Gamma^3}{3}\rho\partial_x\rho \;+\; \mathcal{O}\!\left(\Gamma^5\right),\tag{20}$$

which corresponds to an active stress contribution $m R_1 \Gamma^3 \rho^2/3$. The total active stress is thus given by,

$$\sigma = s\rho(\rho - \rho_0),\tag{21}$$

with $s = -m R_1 \Gamma^3/3$ and $\rho_0 = -2A_1/(R_1\Gamma)$. Together with **Equations (7,11,21)** are the equations of motions of our system.

## Scaling analysis of the equations of motion

We asked how the characteristic time of contractions scales as a function of the width $W_0$ of the confining chamber, according to our theory. For this, we rewrite the equations of motion, **Equations (7,11)**, in dimensionless form

$$\delta^2\partial\hat{x}^2\hat{v} - \hat{v} = \partial\hat{x}(\hat{p}(\hat{p}-1))\tag{22}$$

$$\partial_{\hat{t}}\hat{\rho} = -\partial_{\hat{x}}\hat{\rho}\hat{v}\tag{23}$$

where $\hat{x} = x/W_0$, $\hat{v} = vT/W_0$, $\delta = \ell/W_0$, $\ell = \sqrt{\eta/\gamma}$ and $\hat{\rho} = \rho/\rho_0$ and $T = \gamma W_0^2/(s\rho_0^2)$. The boundary condition then becomes

$$\left[\partial\hat{x}\hat{v} - \frac{1}{\delta^2}(\hat{p}(\hat{p}-1))\right]_{\hat{x}=\frac{\pm\hat{w}(t)}{2}} = 0\tag{24}$$

with $\hat{w}(t) = W(t)/W_0$. To further simplify our analysis, we move to the 'Lagrangian' frame defined by $\chi = \hat{x}/\left(2\hat{w}(t)\right)$, where the equations of motion become

$$\delta^2\partial_\chi^2\hat{v} - \frac{\hat{w}(t)^2}{4}\hat{v} = \frac{\hat{w}(t)}{2}\partial_\chi(\hat{p}(\hat{p}-1))\tag{25}$$

$$\partial_{\hat{t}}\hat{\rho} = -\frac{2}{\hat{w}(t)}\partial_{\chi}\left(\hat{\rho}\hat{v} - \frac{X\partial_{\hat{t}}\hat{w}(t)}{2}\hat{\rho}\right) - \hat{\rho}\frac{\partial_{\hat{t}}\hat{w}(t)}{\hat{w}(t)} \tag{26}$$

with the boundary conditions

$$\left[\partial_{\chi}\hat{v} - \frac{\hat{w}(t)}{2\delta^2}(\hat{p}(\hat{p}-1))\right]_{\chi=\pm 1} = 0 \tag{27}$$

This system of equations has steady-states for $\hat{\rho} = 1$, $\hat{w}(t) = w$, $\hat{v} = 0$, where $w$ is the final width of the film. We next linearize around this steady state, i.e., choose $\hat{\rho} = 1 + \varepsilon\bar{\rho}$, $\hat{w}(t) = w + \varepsilon\overline{w}$, $\hat{v} = \varepsilon\bar{v}$, where $\varepsilon$ is a small quantity. To linear order the equations of motion then become

$$\delta^2\partial_{\chi}^2\bar{v} - \frac{w^2}{4}\bar{v} = \frac{w}{2}\partial_{\chi}\bar{\rho} \tag{28}$$

$$\partial_{\hat{t}}\bar{\rho} = -\frac{2}{w}\partial_{\chi}\bar{v} \tag{29}$$

and

$$\partial_{\chi}\bar{v} = \frac{w}{2\delta^2}\bar{\rho} \quad \text{at} \quad \chi\pm 1. \tag{30}$$

Using **Equations (28,29)**, we find

$$\left(\delta^2\partial_{\chi}^2 - \frac{w^2}{4}\right)\partial_{\hat{t}}\bar{\rho} = -\partial_{\chi}^2\bar{\rho} \tag{31}$$

and the boundary condition

$$\partial_{\hat{t}}\bar{\rho} = -\frac{1}{\delta^2}\bar{\rho} \quad \text{at} \quad \chi\pm 1. \tag{32}$$

We solve this equation by making the Ansatz $\bar{\rho}(t) = \sum_{k=1}^{\infty}\rho_k(t) \quad \cos \quad \left((2k-1)\frac{\pi}{2}\chi\right) + \rho_0 e^{-t/\delta^2}$, and find

$$A_k\dot{p}_k + B_k\rho_k = C_k e^{-t/\delta^2}, \tag{33}$$

with $A_k = \left(\delta^2\pi^2(2k-1)^2 + w^2\right)/4$, $B_k = \pi^2(2k-1)^2/4$ and $C_k = -\rho_0 w^2(-1)^k/\left(\delta^2\pi(2k-1)\right)$. Thus,

$$\rho_k = \frac{C_k}{B_k/A_k - 1/\delta^2}e^{-t/\delta^2} - K_k e^{-(B_k/A_k)t}, \tag{34}$$

where $K_k$ is an integration constant determined from the initial condition. In the following we shall consider the case $\rho_k(t=0) = 0$, i.e. we start with a uniformly stretched film, for which $K_k = \frac{C_k}{B_k/A_k - 1/\delta^2}$.

To determine the timescale of the width contractions we need to remember the conservation of mass

$$M = \int_{-1}^{1} d\chi \frac{w + \bar{w}}{2}(\rho + \bar{\rho}) \tag{35}$$

which yields

$$\bar{w} = w \int_{-1}^{1} d\chi \bar{\rho}. \tag{36}$$

We determine the timescale $\tau$ of contractions from $T/\tau = -\dot{\bar{w}}/\bar{w}$ and find

$$\tau(t) = \frac{\sum_{k=1}^{\infty} K_k \left( e^{-t/\delta^2} - e^{-(B_k/A_k)t} \right) \frac{4(-1)^{k+1}}{\pi(2k-1)} + 2\rho_0 e^{-t/\delta^2}}{\sum_{k=1}^{\infty} K_k \left( e^{-t/\delta^2}/\delta^2 - e^{-(B_k/A_k)t}(B_k/A_k) \right) \frac{4(-1)^{k+1}}{\pi(2k-1)} + \frac{2\rho_0}{\delta^2} e^{-t/\delta^2}}. \tag{37}$$

Thus, the dynamics is governed by multiple relaxation processes with varying timescales. In particular

$$\frac{1}{T} \lim_{t \longrightarrow 0} \tau(t) = \frac{\delta^2}{(w/2)^2 + 1} \tag{38}$$

and

$$\frac{1}{T} \lim_{t \longrightarrow \infty} \tau(t) = \delta^2 + \frac{w^2}{\pi^2} \tag{39}$$

In the experimental parameter regime, the timescale we measure is presumably intermediate,

$$\tau = \alpha \frac{\eta}{s\rho_0^2} + \beta \frac{\gamma}{s\rho_0^2} W_0^2, \tag{40}$$

where $\alpha$ and $\beta$ are dimensionless quantities which we determine numerically. To obtain $\alpha$ and $\beta$ for a given set of input parameters, we numerically solve *Equations (7,11,13)* and extract the timescale $\tau(W_0)$ for several initial widths. We then fit the results to the functional form of *Equation (40)*.

In the experimental regime, using the parameters for which the theoretical contraction profiles best agree with the numerical one (see *Figure 5B*), we estimate $\alpha \simeq 2.2 \pm 0.05$ and $\beta \simeq 0.085 \pm 0.006$. The error estimates were obtained by sampling $\alpha$ and $\beta$ over a range of input parameters between half and twice the best fit values, and evaluating a standard error on the computed values.

## Numerical treatment

To solve *Equations (7,11,13)* numerically, we discretize the system by representing $\rho$ on an equispaced grid between $x = -W(t)/2$ and $x = W(t)/2$, where $W(t)$ is the instantaneous width of the contracting film. The instantaneous film velocity is determined from *Equation (11)* using a second order finite difference scheme. The boundary conditions $\sigma = 0$ at $x = \pm W(t)/2$, are implemented using an asymmetric second order finite difference stencil, see (*Tornberg and Shelley, 2004*). We determine the time derivative of density using *Equation (26)* with the boundary condition specified in *Equation 27*, which account for the

grid contracting with the width of the film. We time-evolve *Equations (26,13)* using a adaptive second order time stepping provided by Scientific Python project (*Jones and Oliphant, 2001*).

