## [Decision Letter]

Thank you for submitting your work entitled "Active Contraction of Microtubule Networks" for consideration by *eLife*. Your article has been reviewed by three peer reviewers, and the evaluation has been overseen by a Reviewing Editor and Naama Barkai as the Senior Editor. One of the three reviewers, Gijsje Koenderink, has agreed to reveal her identity.

The reviewers have discussed the reviews with one another and the Reviewing editor has drafted this decision to help you prepare a revised submission.

Summary:

This study describes the macroscopic contraction of the microtubule network induced in *Xenopus* egg extracts by the addition of Taxol. It was shown before that with these conditions, the motor dynein induce the formation of asters (radial structures) that have a size that is commensurate with the length of the microtubules induced by Taxol. In this study, the larger scale is considered for the first time, and it is shown that the asters will coalesce, inducing the whole network to contract. This is reminiscent of the actin-gel contraction that is observed in the same system, but with a network of microtubules. The observations are interesting and fairly systematic. The quantitative nature of the experimental data allows for the development of phenomenological theoretical models that can describe the data. The message is clear and the article will be a worthwhile addition to the literature, in particular because it shows that contraction is not a unique feature of the actin world, as it occurs also in microtubule networks.

Essential revisions:

1) The claim "Our results suggest that the dynein-driven clustering of microtubule minus ends causes both aster formation and network contraction" (Introduction) is not sufficiently well supported by the data. Firstly, the aster nature of the structures is only weakly supported by the data (the data being shown in Figure 1; especially panel 1C is unconvincing); it would be more convincing to visualize the MT minus ends or the motors. Secondly, the claim that dynein drives minus-end clustering is not well-supported since dynein inhibition only slows contraction, but does not stop it. Is the inhibition not 100% effective, or are other processes involved in contraction?

2) The biological relevance of the work must be made more explicit. The authors claim that the observation of dynein-driven clustering of microtubule minus ends has "strong implications for understanding the role of dynein in spindle assembly and pole formation." It is, however, not clear specifically how the insights obtained here can be translated to those situations. The statement in the Results section "suggesting that dynein is performing the same activity in spindles as in the contracting networks" is particularly unclear.

3) To develop the model (Appendix, “Active stresses from dynein-mediated microtubule interactions), the authors consider point-like objects in 1D, which are said to represent the "minus ends of microtubules". However the attractive interaction between these points extends symmetrically left and right, as if the microtubules were not orientated. The effective steric interaction between microtubules is also only dependent on the position of the minus-end, ignoring the fact that it is really the polymer mass that would induce steric forces, and not just the minus-ends. The plus-ends would be equally relevant. In other words, it seems in this theory that the minus-end located in the middle of the filament. To avoid this inconsistency with the accepted molecular picture, the authors should consider reformulating the model as a theory of many asters coming together. The basic unit would not be a microtubule, but a bunch of them already connected at their minus-ends. The variables 'Xi' could represent the positions of the centres of these asters, and assuming that they are symmetric, it would be justified to use the functional form of the interactions that are surmised. Doing so will not require changing the algebra in the theory, and rewriting its description would be sufficient.

4) The authors briefly mention that networks confined within 10-micron thick chambers do not exhibit any macroscopic contraction. Is there a way to understand this experimental observation and can it be accounted for by the theoretical model?

5) It seems that the contraction is a two-step process in which microtubules first form localized aster. Subsequently, these asters interact with each other in order to form a bulk contractile gel. Their proposed mechanism of end motor clustering explains local contractions that lead to formation of asters. However, presumably once the asters are formed all the microtubule ends are localized at the center of asters. In view of this information, please explain better why interacting asters would form a bulk contractile structure.

6) Previous work by Leibler, Surrey and Nedelec has investigated the behavior of purified systems of molecular motors and microtubules. They have not described bulk contraction but only formation of aster-like structures, which perhaps could be considered to be localized contractions. It would be good to explain the differences between the two analyzed systems and discuss whether the comparison between the two studies might suggest a way to assemble bulk microtubule contractile networks from purified components.

7) Model and data description:

A) Regarding the kinetics of contraction, several important aspects are unclear: what defines t=0? What is the origin of the lag time? And what process actually triggers constriction? A more detailed description of the kinetics of microtubule assembly versus that of motor-driven remodeling would be helpful.

B) It is not evident why the width *W(t)* is used as a read-out for the densification. It seems that fluorescence intensity would be a much more direct measure of concentration. In particular, since the ε does not take into account the thickness of the gel. xz/yz-projections could help to justify this choice.

C) The authors should show fits of ε(t) in Figure 1 and in particular in Figure 5, and comment on the adequacy of exponential fits.

D) Viscoelastic and poroelastic models can be nicely ruled out by showing the dependence of τ(W_0_). The new model clearly is more consistent with the data, but actually does not reproduce the trend very well. The authors should critically comment on this, as well as on other aspects of their model.

E) The model derivation (assumptions, *s*, ρ_0_) should be better explained in the main text to allow the reader to follow the reasoning.

F) The authors should provide a more detailed explanation of what s and ρ_0_ depend on, and give a simple estimate of physically reasonable values. In general, the authors should comment on the connection between the parameters in the continuum theory with microscopic parameters. As it stands, the best-fit parameter values (Results) are meaningless.

G) It is unclear how alpha and beta are derived. What are the input parameters? (Please see the subsection “Scaling analysis of the equations of motion”).

H) Video 1 as well as Figure 1 show very inhomogeneous density distributions. How is this taken into account in the theory or why can it be neglected?

I) The cytochalasin D concentration should be specified.

J) For the PIV analysis, the settings should be specified.

K) In Figure 5: is the initial systematic decrease of ε_∞_a real effect?

8) What are the implications of this new mechanism for other, more well studied, contractile systems, in particular actin-myosin?

9) The statements on alternative methods of study appear to be a bit superficial. The authors are encouraged to discuss better the shortcomings of their approach.

---

## [Author Response]

Essential revisions:

*1. The claim "Our results suggest that the dynein-driven clustering of microtubule minus ends causes both aster formation and network contraction." (Line 94 p4) is not sufficiently well supported by the data. Firstly, the aster nature of the structures is only weakly supported by the data (the data being shown in Figure 1; especially panel 1C is unconvincing); it would be more convincing to visualize the MT minus ends or the motors. Secondly, the claim that dynein drives minus-end clustering is not well-supported since dynein inhibition only slows contraction, but does not stop it. Is the inhibition not 100% effective, or are other processes involved in contraction?*

We thank the reviewers for their comments. In order to address the aster nature of the structures, we repeated the contraction experiment, but with the addition of Alexa Fluor 488 anti-NUMA antibodies. It has previously been shown that NUMA forms a complex with dynein [1] localizes to the cores of Taxol asters in mitotic extracts [2] and has been used as a minus end marker to infer microtubule polarity [3]. Figure 1 was updated to include results showing NUMA localization. Figure 1 illustrates that isolated asters in our system have NUMA at their core as well. As evidence that microtubules in this system can form asters that can fuse together, a basic assumption of our model, Figure 1 and Video 1 were added, directly showing this process for isolated asters. An example field of view for the bulk network is displayed in Figure 1, where tubulin is shown in magenta and NUMA is shown in green. From this image, one can see that NUMA tends to lie in the interior of the structures, arguing that the microtubule minus ends are being focused in the interior, as is the case in asters, providing further evidence of their aster-like nature.

The data presented argue that dynein contributes strongly to the contraction process due to the fact that the contraction timescale can be tuned from a few minutes to well over an hour by inhibiting dynein with varying amounts of p150-CC1. p150-CC1 is a subunit of the dynactin complex that inhibits dynein activity by competing off full length dynactin. Due to this mechanism of action, it remains possible that the entire pool of dynein in the extracts is not being fully inactivated, even with the addition of 2 μM p150-CC1. Thus, it remains possible that the inhibition is not 100% effective and that some small residual pool of active dynein is sufficient to drive contraction, albeit at much slower timescales. A second possibility is that there exists another motor protein that is sufficient to drive contraction in the absence of dynein activity. One possible candidate would be Kinesin-14, a minus end directed motor that can crosslink and slide microtubules, organizes microtubules into asters in purified systems, and is known to be present in *Xenopus* extracts where it has been shown to have a small effect on spindle morphology [4]. However, as the timescale is proportional to 1/s, where s is the strength of the active stress, if we compare the measured timescales between the case with 2 μM p150-CC1 and the case with no added p150-CC1, then it suggests that in the case where dynein is inhibited, the active stress is only ~ 4% of the active stress in the unperturbed case. Thus, even if there is another motor partially involved in the contraction process, ~96% of the stress driving the contraction is being generated by dynein.

In order to clarify this point, the following has been added to the main text.

“We note that even at the highest p150-CC1 concentrations used, the network still undergoes a bulk contraction. This could possibly be due to incomplete inhibition of dynein by p150-CC1, or by another motor protein present in the extract that also contributes to the contraction process. As the characteristic time, τ ∝ 1/s, by comparing the characteristic times in the uninhibited and 2 μM p150-CC1 cases, we can estimate that the strength of the active stress, s, in the 2 μM p150-CC1 condition is only ≈ 4% of the strength of the active stress in the uninhibited case, arguing that even if another motor is involved in the contraction, dynein contributes ≈ 96% of the active stress.”

*2.The biological relevance of the work must be made more explicit. The authors claim that the observation of dynein-driven clustering of microtubule minus ends has "strong implications for understanding the role of dynein in spindle assembly and pole formation." It is, however, not clear specifically how the insights obtained here can be translated to those situations. The statement on p10 lines 259 "suggesting that dynein is performing the same activity in spindles as in the contracting networks" is particularly unclear.*

A substantial amount is known about the behavior of dynein, both from its behavior in purified systems [5] as well as in spindles [6]. Much of the previous work has focused on dynein’s ability to slide microtubules and cluster minus ends together, leading to either aster or spindle pole formation. Models based on this view, e.g. so called “slide-and-cluster models” [7] have had some success describing spindle pole formation, but are essentially one dimensional models and consequently cannot address aspects of the spindle pole formation process such as the “pinching down” of the poles. In this work, we showed how the behavior of individual filaments leads to emergent contractile stresses, and based on our data we think that dynein could be contributing contractile stresses in spindles. By thinking about motor activity in terms of stresses, one has a framework from which to consider more macroscopic aspects of spindle pole formation. For example, in our system, inhibiting dynein leads to a lower contractile stress. Applying this idea to spindles naturally explains why spindle poles unfocus in the absence of dynein activity. Furthermore, the form of the active stress presented here naturally gives insight into how dynein activity regulates the density of microtubules.

To clarify the statement on “dynein performing the same activity” on p10 line 259, the relevant sentence has been changed as follows:

“Fitting a sigmoid function to the τ vs. p150-CC1 concentration curve yields an EC50 value of 0.22 ±. 02 μM (mean ± standard error), similar to the value of ≈ 0.3 μM reported for the effect of p150-CC1 on spindle length in Xenopus extracts (Gaetz and Kapoor, 2004), which is consistent with active stress generated by dynein being required for pole focusing.”

3. To develop the model (5.1.2), the authors consider point-like objects in 1D, which are said to represent the "minus ends of microtubules". However the attractive interaction between these points extend symmetrically left and right, as if the microtubules were not orientated. The effective steric interaction between microtubules is also only dependent on the position of the minus-end, ignoring the fact that it is really the polymer mass that would induce steric forces, and not just the minus-ends. The plus-ends would be equally relevant. In other words, it seems in this theory that the minus-end located in the middle of the filament. To avoid this inconsistency with the accepted molecular picture, the authors should consider reformulating the model as a theory of many asters coming together. The basic unit would not be a microtubule, but an bunch of them already connected at their minus-ends. The variables 'Xi' could represent the positions of the centres of these asters, and assuming that they are symmetric, it would be justified to use the functional form of the interactions that are surmised. Doing so will not require changing the algebra in the theory, and rewriting its description would be sufficient.

The referee is correct. Our microscopic model considers all orientations of microtubules to occur with equal likelihood. This choice is consistent with experimental observations, which do not show strong alignment occurring during contractions. We assume that interactions are statistically isotropic, and make the further simplifying abstraction that steric interactions can be usefully captured by only accounting for MT minus-ends. These assumptions yield a simple model that accounts for the experimental observations.

As the referee correctly points out, our model could be rephrased in terms of higher order fundamental units such as MT-asters, without changing the algebra. We could then (consistently with experiments) postulate that asters stay circular over the contraction process, such that aster-aster interactions occur with equal likelihood in all directions. The aster based formulation, is equivalent to the MT-minus end formulation in all aspects, including the isotropy assumptions required.

In the revised version of the manuscript we chose to keep our explanation of the origin of contractions formulated in terms of MT minus ends as the fundamental units, since this framework allows to explicitly name the microscopic processes that we think are central to this system.

To address the referee's point, we now explicitly state that under the assumption of orientational isotropy, the effects of relative orientations of microtubules average out. Furthermore, we added a sentence in the main text stating the equivalence between our MT-minus end based description and a description constructed by tracking aster cores.

*4. It seems that the contraction is a two-step process in which microtubules first form localized aster. Subsequently, these asters interact with each other in order to form a bulk contractile gel. Their proposed mechanism of end motor clustering explains local contractions that lead to formation of asters. However, presumably once the asters are formed all the microtubule ends are localized at the center of asters. In view of this information, please explain better why interacting asters would form a bulk contractile structure.*

Our interpretation is that microtubules can extend from one aster into the the core of a neighboring aster where forces can be exerted that drive the aster cores together (see Video 1). If there are many asters all trying to coalesce with their neighbors, then the stress on the entire aster network will necessarily be net contractile. However, we don’t think that this process occurs in two discreet steps, where first asters are made and then the asters interact. Rather, what we think is that both of these processes are happening simultaneously and the asters are beginning to interact and coalesce before they are completely formed.

In order to clarify that neighboring asters can fuse, even if the asters are isolated from the network, Video 1 and Figure 1 have been added showing this process occurring in asters that form outside of the main contracting network.

5. Model and data description:

a. Regarding the kinetics of contraction, several important aspects are unclear: what defines t=0? What is the origin of the lag time? And what process actually triggers constriction? A more detailed description of the kinetics of microtubule assembly versus that of motor-driven remodeling would be helpful.

For each experiment, Taxol is added to the extract before the extract is loaded into the microfluidic channels. The channels are then sealed to prevent evaporation, and then placed on the microscope before imaging begins. t=0 is defined as the time that imaging begins, and it occurs approximately one minute after Taxol is added to the extract. The lag time includes contributions from many components, including the time it takes between Taxol addition and the beginning of imaging, the time it takes for microtubules to assemble, and the time for the microtubule network to percolate and remodel. Other than the addition of Taxol to the extract, there is no process that we do that actually triggers the constriction; it happens spontaneously.

To better explain this, to section 4.4 Bulk Contraction Assay the following text has been added.

“t=0 is defined when the imaging begins, ≈ 1 minute after Taxol addition to the extract. After a brief lag time, the microtubule networks spontaneously begin contraction”

*b. It is not evident why the width W(t) is used as a read-out for the densification. It seems that fluorescence intensity would be a much more direct measure of concentration. In particular, since the epsilon does not take into account the thickness of the gel. xz/yz-projections could help to justify this choice.*

We consider both the width W(t) (Figure 2,Figure 3,Figure 6) as well as the fluorescence intensity (Figure 5) as read-outs for densification, and model parameters extracted from fitting the characteristic time scaling with channel width found from fitting W(t) are in agreement with the parameters extracted from the density profile fitting in Figure 5, arguing that there is no systematic bias in choosing one readout over the other. In addition, our value for the final network density, ρ_0_, comes from measurements based on using fluorescence in the network as a proxy for tubulin concentration.

*c. The authors should show fits of epsilon(t) in Figure 1 and in particular in Figure 5, and comment on the adequacy of exponential fits.*

One example fit of epsilon(t) was shown in the inset to Figure 1, though it was difficult to see both the data and the fit due to the size of the lines used. In order to make it more clear, the thickness of the data line was increased. Fits of epsilon(t) for all of the data previously in Figure 1 are now included as Figure 2—figure supplement 1 and as a representative of the data previously shown in Figure 5, fits of epsilon(t) for the 2 μM p150-CC1 data are included as Figure 6—figure supplement 2. In all cases, the data are adequately described by the exponential fits.

*d.Viscoelastic and poroelastic models can be nicely ruled out by showing the dependence of tau(W_o). The new model clearly is more consistent with the data, but actually does not reproduce the trend very well. The authors should critically comment on this, as well as on other aspects of their model.*

We agree that the fit to the tau vs W_0 data does not perfectly reproduce the data, but only the general trend. This could be due to several reasons, including the relative simplicity of our model. It would be possible to extend the model so that the fit is improved, but at the cost of additional model complexity. With the simplicity of the model taken into account, we find the close agreement between the parameters found by fitting the tau vs W_0 data and the parameters extracted from the density profile fits to be remarkable.

*e. The model derivation (assumptions, s, Rho0) should be better explained in the main text to allow the reader to follow the reasoning.*

We agree, and followed the referees advice. The discussion of the model in the main text has been expanded and enhanced.

*f. The authors should provide a more detailed explanation of what s and Rho0 depend on, and give a simple estimate of physically reasonable values. In general, the authors should comment on the connection between the parameters in the continuum theory with microscopic parameters. As it stands, the best-fit parameter values (e.g. bottom p8 and p9) are meaningless.*

ρ_0_ represents the microtubule density at which the dynein driven attraction between minus ends balances the steric repulsion between filaments, and is presumably set by the packing of microtubules. In Figure 6, we show ε_∞_ and hence ρ_0_ to be independent of the total concentration of motor proteins. We also provided an estimate for ρ_0_, but were not clear on this point. To clarify that 30 μM is an estimate for ρ_0_, the text was modified as follows.

“By using fluorescence intensity as a proxy for tubulin concentration (see Materials and Methods), we estimate the final concentration of tubulin in the network to be ρ_0_ ≈ 30 μM.”

To estimate the magnitude of the active stress, we begin with our estimate of ρ_0_ = 30 μM = 3 x 10^-20^ mol/μm^3^. We then estimate the average length of a microtubule to be ≈ 6.5 μm. Since microtubules have 1625 heterodimers per μm [8] we can estimate that there are ≈ 1.8 x 10-^20^ mol/MT. Combining these two estimates leads to a microtubule density of ≈ 1.7 MT/μm^3^. We note that this is also the density of microtubule minus ends. If there are γ dynein per microtubule minus end, each dynein exerts an average force of ≈ 1 pN, and the characteristic interaction length is the average length of a microtubule ≈ 6.5 μm, we can estimate the dipole moment per microtubule to be 6.5 γ pN⋅μm/MT. Combining this with our microtubule density measurement leads to an estimate of the active stress, sρ_0_^2^ ≈ 11 γ Pa. As an estimate of the viscosity of the microtubule network, we take a value measured in spindles in *Xenopus* extracts, η ≈ 2 x 10^2^ Pa⋅s [9]. Combining this with the value of η / sρ_0_^2^ = 0.82 min we measure, gives an estimate of sρ_0_^2^ ≈ 4 Pa. Combining this with our stress estimate derived above from microscopic considerations gives an estimate of γ ≈ 0.4 motors per minus end. Thus, our measured timescale is consistent with our measured microtubule density given an average force per motor of ≈ 1pN and 0.4 dynein per minus end. To provide an estimate of the total active stress as well as things it will depend on, the following has been added to the main text.

“Combining this measurement with an estimate for the network viscosity taken from measurements in spindles of η ≈ 2 x 10^2^ Pa⋅s (Shimamoto et al., 2011), we can estimate the dynein generated active stress to be sρ_0_^2^ ≈ 4 Pa which is consistent with having ≈ 0.4 dynein per microtubule minus end each exerting an average force of 1 pN (Nicholas et al., 2015).”

*g. It is unclear how alpha and beta are derived, what are the input parameters? (see p22)*

In principle alpha and beta can depend on dimensionless combinations of all model parameters and thus need to be determined numerically. To obtain alpha and beta for a given set of input parameters we numerically evaluate the time scale τ(W_0_) for several initial widths and fit the results to Appendix Eqn 40.

To estimate for alpha and beta in the experimental regime, we sampled a large number of sets of the input parameters around the best fit values for which best agreement between experimental and theoretical contraction dynamics was found (Figure 5), varying all parameters by a factor of up to 4. We find that throughout this large region of parameter space alpha and beta vary only weakly and report their mean value and standard deviation.

We thank the referee for pointing out our lack of clarity and we now better explain this procedure in the Appendix.

h. Video 1 as well as Figure 1 show very inhomogeneous density distributions. How is this taken into account in the theory or why can it be neglected?

In this work we approximate the microtubule gel as a smooth continuum, and do not account for the local structure which exists on length scales smaller then typically 100 micron. Such a description is appropriate since, both the size of the system (cms) and the length over which local contractions propagate in the film (η/γ, mm) are much larger then the local structures (10-100 microns).

*i. The cytochalasin D concentration should be specified.*

Cytochalasin D was added to the extracts at a final concentration of 10 μg/mL, as is standard [10]. The concentration is now given in the text.

*j. For the PIV analysis, the settings should be specified.*

Section 4.7 Particle Imaging Velocimetry has been updated to include the settings and algorithm used. The relevant sentence now reads:

“Particle Imaging Velocimetry was performed using PIVLab software (Thielicke and Stamhuis, 2014) using the FFT window deformation algorithm with a 16 pixel interrogation area and 8 pixel step for the first pass and an 8 pixel interrogation area with a 4 pixel step for the second pass.”

*k. In Figure 5: is the initial systematic decrease of epsilon_infinity a real effect?*

While we cannot conclusively demonstrate whether or not the effect is real, we suspect that it is not. Still, the effect is interesting and is an aspect worth following up on. In either case, the main trend that epsilon_infinity does not appreciably vary with dynein inhibition is clearly shown, and is consistent with the proposed theoretical model.

*6. What are the implications of this new mechanism for other, more well-studied, contractile systems, in particular actin-myosin?*

We find the comparisons between the similarities and differences between the contracting microtubule networks shown here and the more well studied contractions in the actin-myosin system to be a fascinating area of consideration. On the microscopic scale, clear differences exist between the two systems, including the orders of magnitude greater persistence length of microtubules relative to actin, and the fundamental differences between myosin-filaments, which can have many myosin heads functioning in parallel, and dynein, which has only two motor domains. Here we present data that argues motor induced minus end clustering leads to active stresses that drive the contraction process. As far as we’re aware, it is not known how myosin interacts with the ends of actin filaments. Thus, the mechanism of end clustering contributing to contractile stresses could be a plausible mechanism for the actin system as well, in addition to mechanisms previously proposed.

To further elaborate on the connection between our system and contractile actin networks the following has been added to the main text.

“Furthermore, the considerations of the model are general, and it will be interesting to consider whether the end clustering mechanism proposed here could contribute to contraction in actin networks as well.”

*7. The statements on alternative methods of study appear quite shallow. The authors are encouraged to discuss better the shortcomings of their approach.*

We view our work as a complementary approach to other methods of study, and there are several other approaches that could be used to study microtubule organization, each with their own advantages and disadvantages. One alternate approach would be to measure stresses in the spindle directly to test whether the net motor generated stresses are contractile or extensile. This approach has the clear advantage of being the most biologically relevant measurement, however it would be extremely challenging technically. Furthermore, stresses in the spindle might reflect the coupled effects of motor activity with contributions from other factors, e.g. microtubule polymerization. Still, it will be interesting to see, at least from a theory perspective, to what extent the dynamics of spindle assembly can be quantitatively understood using a framework based on the dynein induced contractile stress presented here. Another possible approach would be to purify fluorecently labeled dynein and study how it slides pairs of filaments and to measure the behavior of dynein at the filament minus ends. This has the advantage of giving direct insights into how dynein dictates the motion of individual filaments, yet has the disadvantage of the difficulty in relating this microscopic behavior with mesoscale network organization. Our approach has the advantage of directly considering large scale network behaviors, but the disadvantage of top down approaches like the one considered here is that microscopic behaviors of individual components are inferred rather than directly measured. To fully understand the behaviors of a complex system like the one considered here, an integrated approach is needed that combines results considering the system from different perspectives.

**References:**

1. Merdes A, Ramyar K, Vechio JD, Vechio J, Cleveland DW, al E: A complex of NuMA and cytoplasmic dynein is essential for mitotic spindle assembly. *Cell* 1996, **87**:447–458.

2. Gaglio T, Saredi A, Compton DA: NuMA is required for the organization of microtubules into aster-like mitotic arrays. *The Journal of Cell Biology* 1995, **131**:693–708.

3. Mitchison TJ, Nguyen P, Coughlin M, Groen AC: Self-organization of stabilized microtubules by both spindle and midzone mechanisms in Xenopus egg cytosol. *Molecular Biology of the Cell* 2013, **24**:1559–1573.

4. Walczak CE, Verma S, Mitchison TJ: XCTK2: a kinesin-related protein that promotes mitotic spindle assembly in Xenopus laevis egg extracts. *The Journal of Cell Biology* 1997, **136**:859–870.

5. McKenney RJ, Huynh W, Tanenbaum ME, Bhabha G, Vale RD: Activation of cytoplasmic dynein motility by dynactin-cargo adapter complexes. *Science* 2014, **345**:337–341.

6. Elting MW, Hueschen CL, Udy DB, Dumont S: Force on spindle microtubule minus ends moves chromosomes. *The Journal of Cell Biology* 2014, **206**:245–256.

7. Burbank KS, Mitchison TJ, Fisher DS: Slide-and-cluster models for spindle assembly. *Current Biology* 2007, **17**:1373–1383.

8. Waterman-Storer CM, Salmon ED: How Microtubules Get Fluorescent Speckles. *Biophysj* 1998, **75**:2059–2069.

9. Shimamoto Y, Maeda YT, Ishiwata S, Libchaber AJ, Kapoor TM: Insights into the micromechanical properties of the metaphase spindle. *Cell* 2011, **145**:1062–1074.

10. Hannak E, Heald R: Investigating mitotic spindle assembly and function in vitro using Xenopus laevis egg extracts. *Nat Protoc* 2006, **1**:2305–2314.